# Influence of Wearing Ballistic Vests on Physical Performance of Danish Police Officers: A Cross-Over Study

**DOI:** 10.3390/s21051795

**Published:** 2021-03-05

**Authors:** Henrik Koblauch, Mette K. Zebis, Mikkel H. Jacobsen, Bjarki T. Haraldsson, Klaus P. Klinge, Tine Alkjær, Jesper Bencke, Lars L. Andersen

**Affiliations:** 1Department of Physiotherapy, University College Copenhagen, 2200 Copenhagen N, Denmark; mzeb@kp.dk (M.K.Z.); bjth@kp.dk (B.T.H.); 2Department of Occupational Therapy, University College Copenhagen, 2200 Copenhagen N, Denmark; mhyj@kp.dk; 3Danish National Police, 1780 Copenhagen V, Denmark; KPK003@politi.dk; 4Department of Biomedical Sciences, University of Copenhagen, 2200 Copenhagen N, Denmark; talkjaer@sund.ku.dk; 5The Parker Institute, Bispebjerg and Frederiksberg Hospital, 2000 Frederiksberg, Denmark; 6Human Movement Analysis Laboratory, Amager-Hvidovre Hospital, 2650 Hvidovre, Denmark; jesper.bencke@regionh.dk; 7National Research Centre for Working Environment, 2100 Copenhagen Ø, Denmark; LLA@nfa.dk

**Keywords:** police officers, ballistic vest, EMG, lumbar spine, patrol vehicles, muscle endurance

## Abstract

Purpose: We aimed to investigate the influence of wearing a ballistic vest on physical performance in police officers. Methods: We performed a cross-over study to investigate the influence of wearing a ballistic vest on reaction and response time, lumbar muscle endurance and police vehicle entry and exit times. Reaction and response time was based on a perturbation setup where the officers’ pelvises were fixed and EMG of lumbar and abdominal muscles was recorded. We used a modified Biering–Sørensen test to assess the lumbar muscle endurance and measured duration of entry and exit maneuvers in a variety of standard-issue police cars. Results: There was a significant difference of 24% in the lumbar muscle endurance test (no vest: 151 s vs. vest: 117 s), and the police officers experienced higher physical fatigue after the test when wearing a vest. Furthermore, officers took longer to both enter and exit police cars when wearing a vest (range: 0.24–0.56 s) depending on the model of the vehicle. There were no significant differences in reaction and response times between the test conditions (with/without vest). Discussion and Conclusion: Wearing of a ballistic vest significantly influenced the speed of movement in entry and exit of police cars and lumbar muscle endurance, although it does not seem to affect reaction or response times. The ballistic vest seems to impair performance of tasks that require maximal effort, which calls for better designs of such vests.

## 1. Background

Use of ballistic vests is increasingly common among Danish police, military and security personnel on regular duty, due to an increased threat level.

Following a terror strike in Copenhagen, Denmark in 2015, it has become mandatory to wear ballistic vests for Danish police officers when they are on active patrol in metropolitan areas. The vest is bulky, stiff and secured tightly to the trunk with a system of Velcro-straps, and as such could plausibly impact the biomechanics of the upper body negatively. In addition, the added weight has been reported to result in limited mobility and can cause significant discomfort [1,2,3,4,5,6]. The mass of the ballistic vest worn by the Danish police officers ranges 3.5–8.5 kg depending on the units the officers are assigned to. A Swedish study [7] found an increase in musculoskeletal pain among police officers wearing ballistic vests and an increase in sick leave among those officers reporting musculoskeletal pain. There is thus good reason to further investigate the discomfort and added load resulting from use of a vest, as this is likely to influence both welfare and resource management in organizations with personnel whose tasks frequently require wearing of ballistic gear. Several other papers on discomfort related to wearing a ballistic vest have been published [1,2,4,5,8]. Larsen et al. [3] reported an association between discomfort from wearing mandatory equipment and multi-site musculoskeletal pain potentially leading to increased absence from work. Furthermore, Ramstrand and Larsen [7] reported that police officers found that the use of ballistic vests increased physical discomfort. The added mass on the trunk introduced by the vest may result in increased muscle activity and/or strain of the non-contractile tissue to keep the body in balance, which may at least partially explain the increase of the reported discomfort among police officers wearing the ballistic vest [9,10].

Reduced mobility and operational capability, as well as increased prevalence of musculoskeletal injuries, have also been reported [5,6,11,12,13]. However, the knowledge of the impact of ballistic vests on motor control and biomechanics of personnel with tasks similar to those of on-duty police officers is sparse. A study by Ricciardi et al. [12] on military personnel who performed physical tests found that there was no effect of sex and adiposity on a range of physiological performance parameters while wearing a ballistic vest. There are few reports on the impact of wearing a ballistic vest on kinematics and kinetics during gait [2], but no reports on the influence of the vest on motor control.

It is likely that the different attributes of the vest, such as mass, shape, size and how it is fixed to the body, can influence the biomechanics and the ability to activate muscles and regain position after perturbations.

The aim of the present study was to investigate whether wearing a ballistic vest has a significant impact on biomechanics, movement patterns and strategies, endurance of the muscles of the back and time to enter and exit patrol cars in a population of Danish police officers. We hypothesized that the vest would have a significant negative impact on all tasks.

## 2. Methods

### 2.1. Design

The study was designed as a same-day cross-over study where the participants served as their own controls and completed all tests both with and without the vest. The order of vest/no vest conditions in each test was randomized, using computer randomization based on the Fischer–Yates algorithm. If a participant was randomized to, e.g., the “vest first condition”, this participant would perform tests with the vest first and subsequently without the vest.

The study was approved by the local ethics committee (journal number: H-19059765) and all participants gave written informed consent prior to participation.

### 2.2. Participants and Equipment

Forty-one active police officers (5 females) participated in the study. Their characteristics are displayed in Table 1. Out of the total 41 participants, 11 of them belonged to a special response team unit and wore a tactical vest, which was larger and heavier (3.4 kg heavier than the standard vest). The participants wore either their own vest or a vest that was identical to their personally assigned vest. The average weight of the vest in percent of body weight is displayed in Table 1. All tests were performed with ballistic inserts both front and back of the vest at all times, which is the recommended setup for Danish police officers in metropolitan areas: ballistic inserts are made of a composite material that provides extra protection from long-range weapons, whereas the vest itself without the inserts provides protection against small firearms and blade weapons only. However, the ballistic inserts do affect the total weight, stiffness, pressure on the trunk and comfort level of the vest significantly. Body mass, body fat percentage and total muscle mass were obtained with an InBody 770 bioelectric impedance scale (InBody, Los Angeles, CA, USA).

### 2.3. Electromyography

Electromyographic (EMG) signals were recorded with a wireless system (Myon, Cometa, Italy) and sampled in EMGandMotionTools (Cometa, Italy). All raw EMG signals were recorded at 2000 Hz and band-pass filtered at 2–500 Hz. For the amplitude analysis, data were rectified and low-pass filtered at 10 Hz to produce a linear envelope (Figure 1).

The frequency analysis was performed on the raw band-pass filtered data.

EMG was recorded from m. erector spinae bilaterally. We derived EMG using a bipolar electrode setup with Ambu Blue Sensor N-00-S/25 (Ambu, Ballerup, Denmark) electrodes. The electrodes were placed anatomically according to Perrotti [14] with an interelectrode distance of 2 cm in the muscle fiber direction. Prior to electrode attachment, the skin was carefully prepared by removing hair and dead skin using a razor and fine sandpaper and subsequently wiped with an alcohol swab to remove skin lipids and reduce impedance.

All participants performed MVCs for trunk extension, during which the maximum EMG amplitudes (maxEMG) were recorded. All trials were done with participants securely strapped to a bench in a prone position.

For the perturbation test, the participants wore an inertial measurement unit (IMU) to measure movement of the trunk. The IMU was placed so the topmost edge of the IMU lined up with the lowest point on the arch of incisura jugularis sterni. Data from the IMU were sampled along with the EMG data in EMGandMotionTool (Cometa, Italy).

### 2.4. Perturbation

The setup for the perturbation test was a modified version of the setup used by Cholewicki et al. [15]. The participants were semi-seated in the apparatus (Figure 2, left), which constrained pelvic and lower limb movement but allowed movement of the trunk. The participants wore an upper body harness that was loaded with weight equal to 10% of participant body weight (BW) and pulled the trunk backwards from a point between the scapulae, with the resulting force vector perpendicular to the trunk midline. The participants were asked to maintain an upright position, and the vertical position of the trunk was controlled with a laser level. The weight was suspended from a guided cable connected to a remote-controlled electromagnetic device, which released the weight at a random time point within 5 s after activation of the remote. The electromagnetic device was coupled with the wireless EMG system and set a marker in data at the time of magnet release that permitted calculation of reaction time, response time and mean and peak EMG amplitudes. Participants wore an IMU sensor to measure movement of the trunk in the anterior–posterior direction (Figure 3). All participants completed ten perturbation test trials, a set of five with vest and a set of five without vest, with the order of the sets randomized, a 30 s interval between trials and a 2 min interval between sets.

Reaction time was defined as the time from the magnet release to the onset of muscle activity in the m. erector spinae. This was done in accordance with procedures used in previous work investigating reaction time [15,16,17]. We defined the limit for onset of muscle activity to 5% of maxEMG. We checked the sensitivity of this limit against the data. A change from 5% maxEMG to 3% maxEMG only changed the onset time by 2–4 ms, which was roughly 2–5% based on reactions times from previous studies [15,16]. Conversely, setting the level at 5% maxEMG resulted in fewer false onsets due to artifacts. Therefore, we accepted the increase in onset time as negligible and decided to use 5% maxEMG as onset limit.

The response time was defined as the time from magnet release until the participant reassumed the starting position. Stop time was defined as the time point when the standard deviation (SD) of the acceleration in the anterior-posterior direction after the magnet release was within ±10% of the acceleration SD in the 15 ms prior to the magnet release (Figure 4). The mean and peak EMG values were calculated for the first second after the magnet drop.

We calculated the response time to mimic real world events, e.g., a push by an external force. Thus, the response time was a proxy for the officers’ ability to adapt to upper body perturbations caused by an external force.

From a mechanical point-of-view, the increased mass on the trunk alone should not influence the ability to coordinate the movements of the upper body if the same absolute external load is applied. However, we do not know if the influence of the vest’s shape, size, stiffness and fixation to the body could play a role in the ability to coordinate movements of the trunk adequately. We, therefore, expected to find a difference in response time between the conditions—not due to the increased mass on the trunk, but rather due to the configuration and nature of the vest.

### 2.5. Endurance

For the lumbar muscle endurance capacity test, we used the Biering–Sørensen test [18], which is a widely used antigravity test [18,19,20,21,22]. This test has proven to be a good indicator for the development of low back pain [19]. All participants were instructed to assume a prone position on a bench with their anterior superior iliac spines just short of the edge of the bench, such that the pelvis was firmly supported, while the trunk was free to move. A cylindrical support was placed beneath their ankles, and the lower extremities were fixed firmly to the table with three strong straps: One strap was secured across the ankles, one just proximal to the knees and one immediately distal to the gluteal muscles (Figure 2, right).

Participants were then instructed, on a signal from the instructor, to raise the trunk from its resting position on the floor, cross their arms across their chest and maintain a horizontal position until exhaustion.

Throughout the test, the level of the trunk was monitored by the examiner, who also gave instructions and encouraged the participant verbally. All participants performed one endurance test with the vest and one without the vest. There was a 5-min interval between the tests, where the participants were asked to eat a piece of fruit or a cereal bar and hydrate as needed.

Furthermore, to confirm fatigue at the end of the test, based on the EMG signal, we calculated the median frequency (MF) from the power spectrum obtained during this test. The MF of the power spectrum was calculated using Fast Fourier Transformation (FFT) on epochs of 1 s in the first 15 s and the last 15 s of the test, using Cometa’s software EMGandMotionTools (Cometa, Italy). To enable comparison of the MF between participants, we calculated the MF from the last 15 s relative to the initial 15 s. Thus, the outcomes from the endurance test were time (seconds) to fatigue and relative median frequency difference (start to finish).

### 2.6. Vehicle Entry and Exit

To investigate real-world influence of the vest on a frequently occurring job-related police task, participants were asked to enter and exit three standard-issue Danish police vehicles under conditions similar to non-emergency patrol duty. Vehicles included a station wagon (VW Passat), an MPV (VW Touran) and a minibus (VW Caravelle), all of which are widely-used patrol cars in the Danish police. The participants entered and exited every car five times with and without the vest and were instructed to perform these actions at a leisurely pace (mimicking normal, non-emergency circumstances).

We captured the entries and exits with a video camera (C920 PRO HD, Logitech) that filmed at 30 Hz. Based on frame-by-frame analysis of the video recordings, we calculated durations of entries and exits for each vehicle, with and without ballistic vests, operationalized as the time elapsed from the participant opened the door until the door was closed after the entry/exit. Start time was defined as being one frame before door-movement was detected, and stop time was set to two frames after the door had fully closed.

### 2.7. Statistical Analysis

We performed all statistical analyses with JMP 13 (SAS institute Inc., Cary, NC, USA). Initially, we used descriptive statistics to determine the central tendency and variation of the participants’ background variables, and all data were accepted for normality by visual inspection of histograms. Student’s *t*-tests were used to test the hypothesis that a difference in the outcomes between the vest/no vest conditions existed. The results are reported as means and standard deviations (SD) unless otherwise stated.

## 3. Results

Figure 5 depicts the comparison between the vest/no vest conditions for the reaction time, response time, mean and peak EMG response, duration of endurance and MF during the endurance test. We found no statistically significant difference between the conditions in reaction time (mean difference: −0.002 s; 95% CI [−0.008; 0.004]), response time (mean difference: 0.03 s; 95% CI [−0.03%; 0.09%]) and mean EMG (mean difference: 0.09% EMGmax; 95% CI [−0.33%; 0.53%]) or peak EMG (mean difference: 2.68% EMGmax; 95% CI [−1.81; 7.17]) response. There was a statistically significant reduction of the mean endurance duration in the vest condition when compared to the no vest condition of 34.1 s (95% CI [7.64 s; 60.5 s]). Furthermore, the MF declined 45% in the vest condition and 49% in the no vest condition, indicating that the participants were physiologically fatigued. The difference in MF between conditions was 4.6 Hz (95% CI [0.8 Hz; 8.5 Hz]) and statistically significant (*p* < 0.001).

Table 2 reports the results observed during entry and exit of two types of standard police vehicles. We found that there were statistically significant differences between the conditions in both entry and exit of the VW Passat and the VW Touran, indicating that, when wearing the vest, the time duration of both entering and exiting the two car types was longer than when the tasks were performed with no vest.

## 4. Discussion

We performed a crossover-study to investigate the influence of wearing a ballistic vest on biomechanical measures in police officers. We found that wearing a vest significantly influenced the duration of endurance and speed of entry and exits from police cars. We found no difference in reaction time, response time or in the EMG activity between the vest and no-vest conditions during the perturbation test.

The reaction times in the present study are markedly longer than what has previously been reported [15,16,17,23,24]: Abboud [17] performed a similar experiment where participants experienced a sudden load increase of the trunk and found a mean reaction time of 94 ms, while the pooled mean reaction time (left and right side pooled) in the present study is 115 ms. Abboud et al. [17] asked their participants to perform an active contraction of the trunk flexor muscles of 10% MVC. The difference in results between their and our study could be partially due to the active contraction priming the sensory system to be more alert, and therefore more ready for the perturbation resulting in shorter reaction times. The difference may also be explained other factors: differences in setup (e.g., exact point of fixation on the pelvis), a larger electromechanical delay in our setup or differences in the verbal instructions given to the participants prior to the magnet release. The difference in reaction times between the present and previous studies does not have significantly impact the conclusions of this study, however, as our main objective was to investigate the effect of wearing a ballistic vest, i.e., determine the relative reaction and response times in the vest/no-vest conditions rather than the absolute reaction and response times of the participants.

The difference in reaction times observed between the vest and no-vest conditions was not statistically significant. This would suggest that the vest does not greatly impact the speed of recruitment of the m. erector spinae, for which reaction time is a direct measure. A change in reaction time would be caused by an altered recruitment pattern to which the wearer of the vest would have to adapt.

We did not observe any difference in response time between the conditions. We expected an increased response time, mean EMG activity and peak EMG activity or at least one of these outcomes due to the increased proprioceptive and tactile stimulus that the vest introduces.

A possible explanation for the lack of difference in response time between the vest and no-vest conditions is most likely that the effect of the sensory stimulus from the vest on the motor system is either not present or simply too small to cause more than a subtle additional stimulus that is below the responsiveness threshold of the test setup. Adding to this, the task was limited to movements in the sagittal plane with a fixed pelvis, which made the task comparatively simple, and, therefore, the differences could have been too small for the EMG to record. Further research on situations closely approximating more complex real-life situations should be carried out to better determine the influence of a ballistic vest on the reaction ability of officers during everyday police work, e.g., research on avoiding thrown objects, doing rapid shifts of direction while moving, etc., while wearing a vest. This would also help to better determine the impact of a ballistic vest on the overall workload experience in connection with patrol duty-task completion.

We found a 23% decrease of the endurance time during the vest condition. This was no surprise, because extra weight is added to the trunk when wearing the vest. Although the weight of the vest is quite small it is added with a relatively long moment arm, which subsequently produces a much larger flexor moment on the lumbar spine for the m. erector spinae to balance.

The decrease in MF over the course of the endurance test emphasizes that the participants were fatigued after the endurance test. The no vest endurance trials were on average 23% longer than vest condition trails which may explain the larger drop in frequency in the no-vest condition.

In the test for entry and exit performance, we found that it took the officers significantly longer to enter and exit both the Passat and the Touran when wearing a vest, whereas there was no difference when entering or exiting the Caravelle regardless of vest condition. The largest difference was found in the Passat. This is most likely due to the lower door opening compared to the Touran and especially Caravelle. In combination with the vest, the lower door opening in the Passat increases the work requirement for the officers, which likely results in the longer time spend to enter and exit the car. Furthermore, there is also less space available inside the car and the officers assumed a more squeezed position in the passenger seat with larger degree of hip and lumbar flexion. This flexion caused the weapons belt to collide with the vest, making the sitting position in the passenger seat rather uncomfortable. Moreover, in the cars with a small door opening, the officers had to lean more forward to exit the car. In this instance, the collision between the weapons belt and vest is also a big issue. These results may be taken into account by decision makers when new vehicle contracts are negotiated.

One limitation of this study is the sample size. We aimed to include 60 participants in the study, but, due to the COVID-19 lockdown in Denmark in March 2020, we were forced to terminate data collection prematurely. This resulted in a sample size of 41 and has caused a loss of power in the statistical analyses. Especially endurance, reaction time and response time are underpowered (0.64, 0.5 and 0.4, respectively). Therefore, results in these outcomes should be interpreted cautiously. However, the effect sizes of reaction and response time (0.002 and 0.03 s) are modest, indicating that the clinical relevance is most likely negligible even if statistically significant differences should exist.

Another limitation is related to the endurance test that had to be carried out twice with only a short break in between tests, providing limited time for restitution, and thus possibly affecting performance on the second endurance test. We contemplated if the “vest first” randomization would exhaust the participants more than the “no-vest first” randomization. However, we did not find any statistically significant effect of randomization order on endurance performance. A strength of this study is that participants used a vest identical or similar to the normal vest that they were accustomed to wear as part of their standard equipment. Thus, there was no influence of a possible learning effect on the results.

Finally, we included a sample which reflected the general population of Danish police officers and consequently the sample was unequal regarding the male–female ratio. We included only five women, and this could potentially have influenced the results. We did find that the women had significantly longer endurance times and significantly larger differences between the vest conditions. However, this difference between conditions could not be explained by sex as it was relative to the overall endurance time.

In conclusion, we found that that the ballistic vest has significant influence on the endurance of the lumbar muscles and on time required of to enter and exit standard-issue police vehicles but shows no significant impact on reaction or response time in a perturbation test, although these results should be interpreted with caution due to our findings being underpowered. Overall, however, given the heavy influence of the vest on endurance and performance in a low-load routine police task, more research into the influence of ballistic vests on officer performance and well-being is warranted, as are further efforts to improve vest designs to minimize their impact on on-duty police personnel.

## Figures and Tables

**Figure 1 sensors-21-01795-f001:**
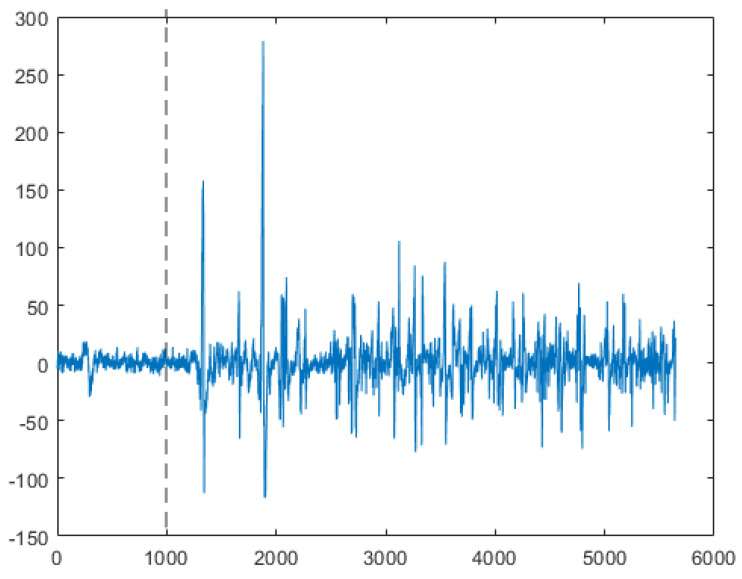
Representative examples of the raw (**top**), rectified (**middle**) and lowpass filtered (**bottom**) EMG signals just before and after the magnet release in the perturbation test. Vertical dashed line illustrates time of magnet release.

**Figure 2 sensors-21-01795-f002:**
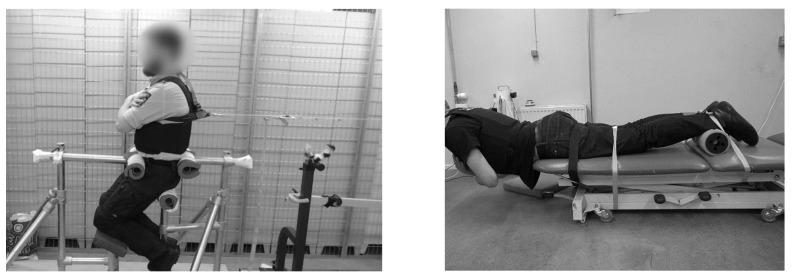
Illustrations of perturbation (**left**) and endurance test (**right**) setups.

**Figure 3 sensors-21-01795-f003:**
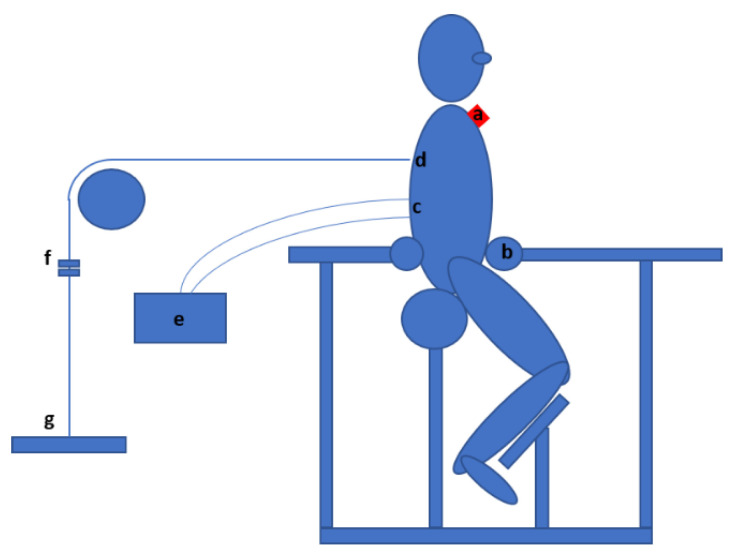
Illustrates the details of the perturbation setup. Letters a–g denote key elements in the setup: (**a**) Inertial Measurement Unit; (**b**) custom made apparatus to constrain hip movement; (**c**) EMG electrodes on m. erector spinae; (**d**) attachment of wire with a harness; (**e**) EMG receiver; (**f**) magnet device; and (**g**) weights equal to 10%BW.

**Figure 4 sensors-21-01795-f004:**
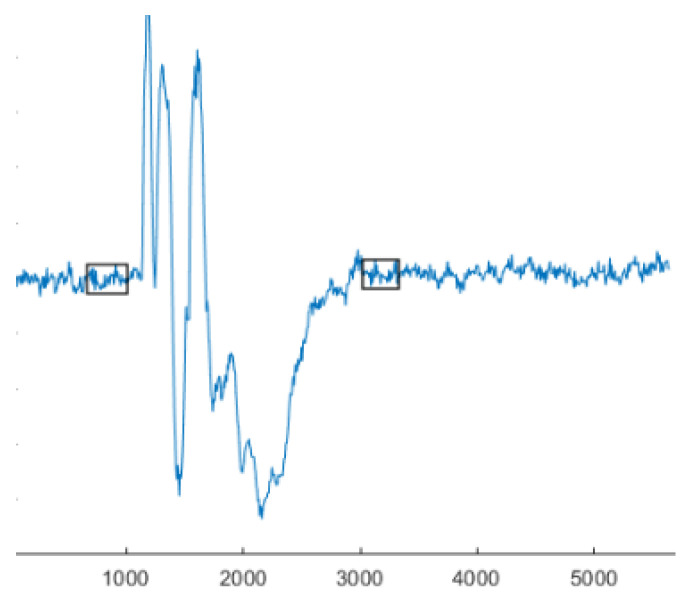
Representative example of acceleration data from the IMU. First rectangle contains the baseline data for steadiness while the second rectangle illustrates the return to steadiness and the stop-time of the response time, as it is the first instant where the SD of the accelerations is within 10% of the baseline acceleration.

**Figure 5 sensors-21-01795-f005:**
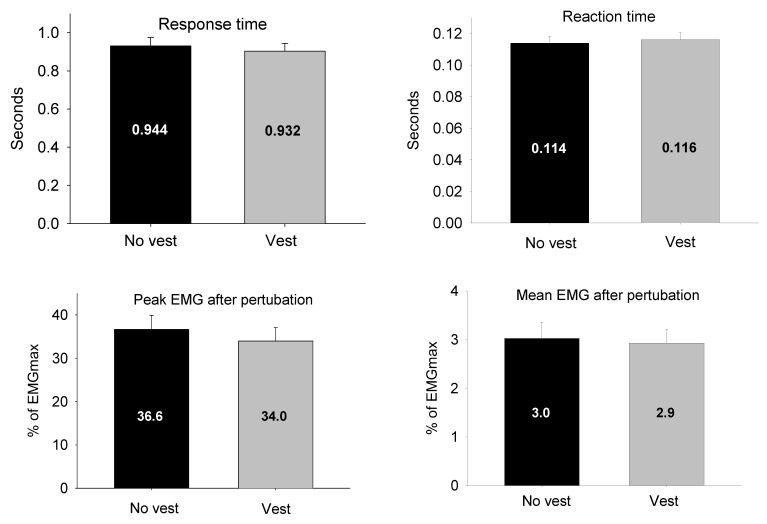
Results from perturbation and endurance test. * denotes statistically significant differences at *p* < 0.05.

**Table 1 sensors-21-01795-t001:** Characteristics of the participating police officers (n = 41).

	Mean (SD)
Sex (F/M)	5/36
Age (years)	34.8 (8.6)
Body weight (kg)	86.2 (14.1)
Height (cm)	179.9 (6.8)
Muscle mass/BW (%)	45.9 (4.1)
Body fat (%)	19.5 (6.8)
Vest/BW (%)	6.4 (1.8)

**Table 2 sensors-21-01795-t002:** Entry and exit times for three different patrol cars with and without vest. * denotes statistically significant differences at *p* < 0.05.

Type of Car	Entry/Exit	Vest/No Vest	Time (s) Mean (95% CI)	Difference (s) Mean (95% CI)
Passat	Entry	No vest	5.0 (4.9; 5.16)	−0.56 (−0.80; −0.33) *
		Vest	5.6 (5.41; 5.79)	
	Exit	No vest	5.4 (5.28; 5.49)	−0.38 (−0.54; −0.22) *
		Vest	5.8 (5.64; 5.9)	
Touran	Entry	No vest	4.6 (4.54; 4.66)	−0.24 (−0.43; −0.06) *
		Vest	4.8 (4.72; 4.88)	
	Exit	No vest	5.1 (5.06; 5.14)	−0.28 (−0.43; −0.14) *
		Vest	5.3 (5.24; 5.36)	
Caravelle	Entry	No vest	4.9 (4.84; 4.96)	−0.07 (−0.24; 0.11)
		Vest	5.0 (4.94; 5.06)	
	Exit	No vest	5.1 (5.05; 5.15)	−0.12 (−0.26; 0.02)
		Vest	5.2 (5.15; 5.25)	

## Data Availability

The data presented in this study are available on request from the corresponding author. The data are not publicly available due to confidentiality restrictions.

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
