# Peer review of "Influence of Wearing Ballistic Vests on Physical Performance of Danish Police Officers: A Cross-Over Study"

_sensors, 2021, doi:10.3390/s21051795_

Round 1

Reviewer 1 Report

The paper describes the impact of ballistic vest on basic activities found in Danish police operations. The research methods are clearly described enough to reproduce the experiments. The result showed that fatigue of wearer and the movement speed in entering and exiting police vehicles are significantly increased with ballistic vest condition while the reaction and response time are not affected by the ballistic vest (but not focused on whether the resultant values are acceptable or not). The paper is concise and well organized.

I have two suggestions to improve the paper.

  • The discussion regarding the activities of entering and exiting police vehicles (Table 2) seems missing in Section 4. It is interesting to understand the reason why the car model affect the entry/exit performance. 
  • Please describe the effect of the skewed gender ratio on the results.

Reviewer 2 Report

In the article “Influence of Wearing Ballistic Vests on Physical Performance of Danish Police Officers: A cross-over study,” Henrik Koblauch et.al have designed a cross-over study to investigate the effect of wearing ballistic vest on reaction and response time, lumbar muscle endurance and police vehicle entry and exit times of Danish police officers. Although the article provides some interesting statistical analyses to elucidate the potential correlations, I feel like the current version of the paper does not fit for the aims and scopes of the journal and is missing lots of data.

  1. The paper provided much more statistical results rather than focusing on how they measure the signals using the sensing system. The authors should provide more detailed figures about how the signals look like, how they filtered the signal (what the filtered signals look like?) and how they built the sensing system. From the current version, the author only showed one figure (Fig. 1) without any labels of the testing system.
  2. Specifically, I think the authors should provide more detailed figures (showing every part of the measuring system, the representative signal data, etc.) rather than elaborating the set-ups of sensing system and data processing in the Methods part from section 2.3 to 2.6. I recommend that in each section the authors provide equipment/instrument set-ups, followed by the original/unfiltered data and then showing the filtered data.
  3. Without the above information mentioned, I feel like it is very hard to review the paper.

Reviewer 3 Report

Dear authors: This is a very interesting topic providing evidence for the measurement of potential detriments of ballistic vests on postural muscle endurance and specific movements in Danish police officers. There seems to be sufficient evidence illustrating the increased weight (although quite small) of the ballistic vests is enough to cause a reduction in postural muscle endurance. Additionally, the cumbersome shapes of the ballistic vests reduce the effectiveness of minor police actions, which in combination may summate to potential career detriments. I was surprised to see there is remarkably little written about your significant results in the discussion section. Most of the discussion section is focused on reaction time, which is understandably the key component and primary focus, however your significant results warrant more than two sentences (in other words, expand on this). The manuscript is amazingly easy to follow as it is very well-written, despite the interestingly complex methodology. This is an interesting area and I commend your group for an excellent first draft. I and am interested to see any longitudinal studies to alleviate this issue.

Round 2

Reviewer 2 Report

I am largely satisfied with the current version. The authors have addressed all my concerns with detailed data explaining how they processed the data and what the system looks like.